# African elephant poaching rates correlate with local poverty, national corruption and global ivory price

Severin Hauenstein [1,2], Mrigesh Kshatriya[3], Julian Blanc [3,4], Carsten F. Dormann [1] & Colin M. Beale [2]

Poaching is contributing to rapid declines in elephant populations across Africa. Following high-profile changes in the political environment, the overall number of illegally killed elephants in Africa seems to be falling, but to evaluate potential conservation interventions we must understand the processes driving poaching rates at local and global scales. Here we show that annual poaching rates in 53 sites strongly correlate with proxies of ivory demand in the main Chinese markets, whereas between-country and between-site variation is strongly associated with indicators of corruption and poverty. Our analysis reveals a recent decline in annual poaching mortality rate from an estimated peak of over 10% in 2011 to <4% in 2017. Based on these findings, we suggest that continued investment in law enforcement could further reduce poaching, but is unlikely to succeed without action that simultaneously reduces ivory demand and tackles corruption and poverty.

[1] Department of Biometry and Environmental System Analysis, University of Freiburg, 79106 Freiburg, Germany. [2] Department of Biology, University of York, YO10 5DD York, UK. [3] United Nations Environment Programme, MIKE – CITES Secretariat, P.O. Box. 30552-00100, Nairobi, Kenya. [4] Wildlife Management Unit, Ecosystems Division, United Nations Environment Programme, P.O. Box. 30552-00100, Nairobi, Kenya. Correspondence and requests for materials should be addressed to S.H. (email: severin.hauenstein@biom.uni-freiburg.de)

Recent high-level political summits[1] and a number of well-publicised ivory destruction events in China[2] and the USA[3] have pushed elephant poaching high up the international agenda. The evidence for an increase in African elephant (*Loxodonta africana*) poaching since the early 2000s is compelling: seizures of illegal ivory shipments have been rising[4], core populations both inside and outside protected areas[5] falling by 30% in seven years[6] and the number of dead elephants found poached have increased[7]. Conservation efforts have focussed on demand reduction (China's recent ban on ivory trade is considered a major success[8,9]) and increased control of supply (e.g. Tanzania's controversial Operation Tokomeza[10] and the call to scale up elephant anti-poaching funds[11]). Figures presented at Convention on International Trade in Endangered Species of Wild Fauna and Flora (CITES) meetings since 2011 suggest poaching rates may be responding, but the effectiveness of different interventions is unclear and empirical data are scarce[12].

Elephants are the very definition of charismatic megafauna, but they are also important engineers of African savannah and forest ecosystems[13] and play a vital role in attracting ecotourism[14], so their conservation is a real concern. While it seems clear that elephant poaching is a pan-African problem[6,7], the effect of poaching is not uniform; some populations are stable or increasing[15] (e.g. South Africa's Kruger National Park) and elephant poaching remains virtually unknown in several African protected areas (e.g. Etosha in Namibia). Such spatial and temporal variation allows us to evaluate the effectiveness of current conservation efforts and identify solutions to the poaching problem.

Current discussion of how to reduce poaching focuses on two areas: reducing demand[16,17], and reducing supply[18]. In recent years, increased demand for ivory in East Asia and particularly China is widely perceived to be the ultimate driver of increased poaching in Africa, primarily based on analysis of the destinations of intercepted ivory shipments[4], the growth in per capita income[19] and the traditional market for ivory in China[20]. Combined with an increasingly large economic involvement of China in Africa, shortening the links between resource and market, Chinese demand has been blamed for rising ivory prices and fuelling the rapid increase in illegal activity[21]. The illegal trade in ivory, however, is complicated: certainly Chinese demand is important, but recently more large seizures of raw ivory were made in Thailand, with transit centres for shipping to other East Asian states such as Malaysia, the Philippines and Vietnam[4]. Reducing demand has been seen as a crucial step in stemming poaching in Africa[18], yet the economics of illegal trade make this complicated[22]: trade bans and associated ivory seizures may even increase poaching incentives by causing price rises in elephant ivory[23].

Supply-side anti-poaching policies focus on increased effectiveness of law enforcement: more patrols by better resourced rangers[24]. In practice, traditional law enforcement activities can reduce illegal activities[24,25]. Unfortunately, as commodity prices rise, law enforcement becomes inadequate with numerous examples of thriving illegal markets persisting for natural (e.g. rhino horn) and other (e.g. narcotics, arms, etc.) products despite high investment in law enforcement[26]. Moreover, endemic corruption and limited capacity in many source countries means that, even if arrests are made in the field, prosecutions may fail or enforcement focuses only on the lowest tier of individuals involved in the trade: effective law enforcement may only be possible if corruption levels are low and enforcement capacity high[25,27]. Additional supply-focused solutions draw on the suggestion that poaching rates may be highest in the poorest regions, where the financial temptations of illegal activities are relatively greater[28]. This has generated interest in community-based conservation programmes that seek to tie conservation improvements directly to poverty alleviation[29] and there is evidence this can reduce local poaching rates[30].

Although it is plausible that elephant poaching cannot be halted without interventions aimed at multiple stages of the ivory trade, it is likely that elephant poaching responds more strongly to certain interventions than others. Here, we have two aims: (a) to test whether local conditions in different sites and in different years can explain variation in poaching rates between and within elephant populations; and (b) to identify the processes to which poaching rates are most sensitive, thereby identifying priority targets for conservation. Based on hierarchical regression models, correlating annual carcass-encounter data from 53 African Monitoring the Illegal Killing of Elephants (MIKE) sites (see Fig. 1) between 2002 and 2017 to local, regional and global socio-economic covariates, we explore the possible contribution of relevant drivers to reducing poaching.

## Results and discussion

**Correlates of elephant poaching.** Both demand and supply could affect ivory poaching rates, so we identified large-scale ivory seizures and mammoth ivory prices in the main Chinese markets as likely proxies for annual variation in ivory demand and used site- and country-level variables such as infant mortality rate (IMR), poverty density (both indices for poverty), corruption perception index (CPI), site area and law enforcement adequacy as proxies for supply-limiting processes (Supplementary Table 1). Using these data we fitted a Bayesian lasso-regulated hierarchical regression model to estimate the annual proportion of all encountered elephant carcasses that were identified as illegally killed (PIKE; see Eq. 1) between 2002 and 2017 in all monitoring sites. For independent validation[31], we fitted the model to training data (2002–13), and compared estimated PIKE for 2014–17 to the respective observed PIKE ($\widetilde{R}^2 = 0.48$, Fig. 2b). We found that annual effects for 2014–17 (including a predicted change in direction of poaching rate) were very similar for the model using training data, compared to a model fitted to all data (Fig. 2a), providing independent support for our model. We estimated observed annual continental poaching rates (black crosses in Fig. 2a) as the sum of all observed carcasses across sites, derived annual continental PIKE values and computed annual continental poaching rates (see Eq. 7). These raw results may be biased downwards, because sites that find more carcasses dominate the continentally aggregated PIKE observations, but they may find more carcasses because they are better resourced and hence tend to have lower poaching rates than sites with fewer observations.

In the regularised model, all covariates, except for site area showed non-zero correlations with PIKE, but only a subset showed credibility intervals (CIs) that excluded zero (Supplementary Table 2). Specifically, we found a strong positive association between ivory price and annual variation in poaching rates (Fig. 3h), while site level variation was positively correlated with poverty density (number of people per km$^2$ earning less than US$ 1.25 per day) and negatively correlated with estimated law enforcement adequacy (Fig. 3d, f). We also found strong evidence that PIKE decreases with falling national corruption (Fig. 3c). While we found strong correlates of PIKE, it is important to note that substantial residual variance remains unexplained (see Fig. 3). We found no evidence that alternative temporal lags in ivory price or seizures changed our results (Supplementary Tables 6–13): an effect of seizure rate at different lags was never supported in our models. Ivory prices with one year lag were correlated with PIKE but with smaller regression coefficients than the zero-lag models, while lags of two years showed no influence on PIKE

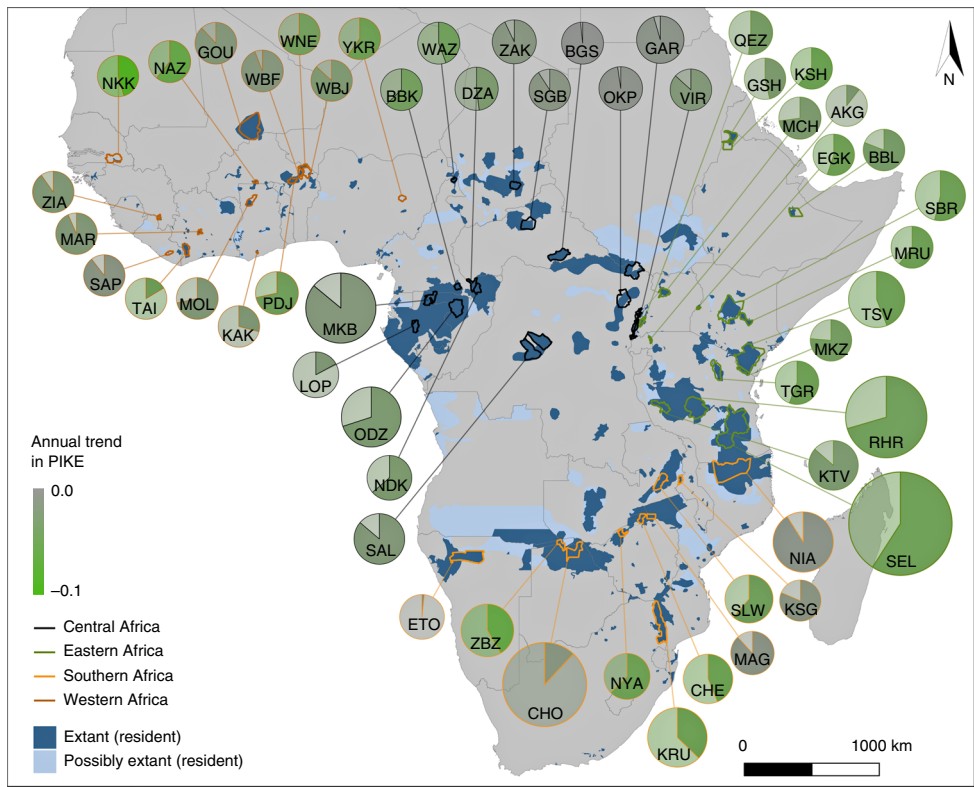

**Fig. 1** Annual poaching trend by site. Map of MIKE sites[62] in Africa[76], showing estimated elephant population sizes[15] (from <10 in Niokolo-Koba National Park, NKK, to 45,254 in Selous-Mikumi Game Reserve and National Park, SEL) proportional to the size of the respective pie chart, estimated median proportion of illegally killed elephants (illegal = solid pie piece, legal = transparent colours), annual trend in estimated PIKE (green = decline [≙ less poaching], grey = no decline) between 2012 and 2017 (for site names see Supplementary Results), and the known and possible range of African savannah elephants[51]

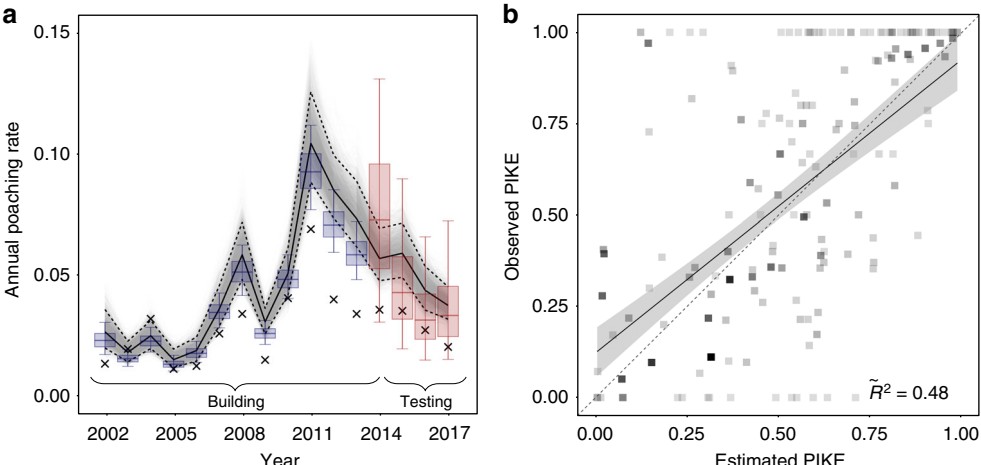

**Fig. 2** Estimated annual poaching intensity—observed and estimated. **a** Annual estimates of per capita poaching rate across 53 MIKE monitoring sites. Displayed estimates are annual median poaching rates across all sites in each year, derived from 3000 MCMC samples. Grey lines display estimates from the model fitted to all data, with solid black line showing median estimates and dashed lines outlining 90% CI. Blue boxes represent estimates from the model (same structure) fitted to training data (2002–13) only, red colour highlights estimates for test data (2014–17). Boxplot centre lines represent median estimates, box bounds first and third quartiles, and whiskers 90% CIs. Crosses represent overall observed poaching rate across all sites and will be biased towards sites where more carcasses are found. **b** Median PIKE estimates for the testing period (2014–17) from 3000 MCMC samples of the model fitted to training data (2002–13) only, compared against the respective observed PIKE values. Darker grey indicates larger sample size. The black line and error envelope represent the mean estimates and the 95% confidence interval from a weighted regression fitted to these data, the dashed line represents identity. $\widetilde{R}^2$ of weighted correlation is 0.48 (90% CI: 0.39–0.54)

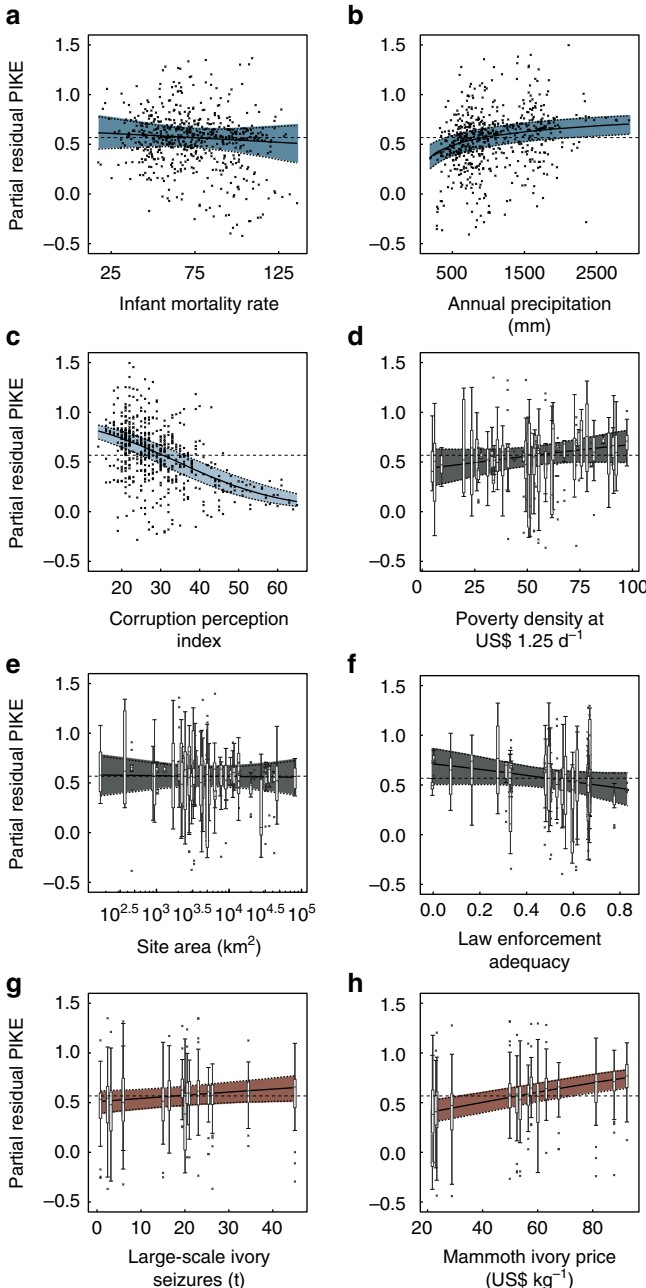

**Fig. 3** Conditional relationships between key covariates and the estimated proportion of illegally killed elephants (PIKE). Site by year covariates (steel blue) **a** infant mortality rate and **b** annual precipitation, the country by year covariate (light blue) **c** corruption perception, the site level covariates (dark grey), **d** poverty density, **e** site area, and **f** law enforcement adequacy, and the annual (brick red) **g** large-scale ivory seizures, and **h** ivory price. Error envelopes represent 90% credibility intervals from 3000 MCMC samples, horizontal dashed lines illustrate the estimated intercept median. All effects plots are overlaid with response-scale partial residuals (points for site and country by year covariates and boxes for site and annual covariates). Boxplot centre lines represent median estimates, box bounds first and third quartiles, and whiskers 90% CIs. All plots are scaled the same to make effect sizes directly comparable

(Supplementary Tables 6–13). This suggests that temporal lags between market prices and poaching rates are themselves short, supporting the evidence that the ivory trade is run by highly organised criminal networks with good knowledge of markets[32].

**Annual poaching rates and trends**. Our estimates of the annual PIKE of the first part of the time series are broadly consistent with estimates from earlier analyses[7,12]. In later parts, we found a peak in 2011 and slowly decreasing PIKE levels thereafter: the median annual poaching rate in MIKE sites in 2011 was 10.4% (90% CI: 8.8%–12.5%), but fell to 3.7% (90% CI: 3.2%–4.5%) in 2017 (see Fig. 2a). Elephant population growth rates can be 5% per year[33,34], suggesting current poaching rates could be sustainable if poaching mortality entirely compensated for natural mortality. Unfortunately, there is little empirical data available to identify the proportion of poaching mortality that may be additive, but because the deaths of larger, older elephants (particularly females) can reduce survival and birth rates among other individuals of the herd[35,36], poaching mortality is likely to be at least partially additive. Consequently, the overall pattern of poaching offers no room for complacency if demand for ivory remains high. Comparison with the case of the Black Rhinoceros (*Diceros bicornis*) offer a cautionary example: a species which had a population of around 100,000 in 1960, yet poaching caused a decline of 97.5% by 1995, and even the most highly protected populations have suffered from severe poaching[37].

Separating poaching rates by region revealed similar temporal patterns among all regions, but large differences in estimated poaching rate (Fig. 4). MIKE sites in western and central Africa recorded much higher overall rates of poaching than eastern and southern populations, confirming a growing literature[38,39]. Temporal differences by region were small, reflecting mainly lower peaks in southern Africa and slight differences in timing of peaks between regions. This likely reflects in part constraints imposed by the model (PIKE must respond similarly to changes in ivory price), but because our model incorporates site- and country-level covariates and random effects, such consistent between-region patterns suggests all regions do show broadly parallel changes. Although savannah elephant populations are largest in eastern and southern Africa, high poaching rates in western and central Africa are a particular concern because these regions are the only homes of already heavily depleted forest elephant populations[39].

**Identifying conservation targets**. To examine the efficacy of feasible intervention targets, we estimated the change in median poaching rate if each variable in turn was set to the most optimistic recorded level for that variable in the actual data (i.e. we explored for example the effect of returning to 2002's ivory prices of US$ 22 kg⁻¹, the consequence of all countries achieving Botswanas 2012 Corruption Perceptions Index (CPI) score, and the adequacy of law enforcement estimated for Namibia's Etosha National Park, etc.). As such, our test of possible effectiveness combines information on how much site level covariates could be improved and the sensitivity of poaching rates to each covariate. The imposed changes represent plausible conservation targets as at least one country, site or year has managed to achieve this value, but predictions should be treated cautiously as they are derived from correlative models. Overall poaching levels seemed sensitive to changes in indices of poverty, but were more strongly linked to corruption and changes in ivory price, whilst seizures and site area showed minimal impacts (Fig. 5). Of course, at different sites and in different countries, different strategies are likely to be effective although we found that virtually all sites would profit the most from country-wide reductions in corruption (see Supplementary Results for site- and country-level evidence). At a regional level we estimated ranked efficacy of different intervention targets to be nearly identical to the overall pattern, the only difference being a slight increase in relative effectiveness of improved law enforcement in western Africa

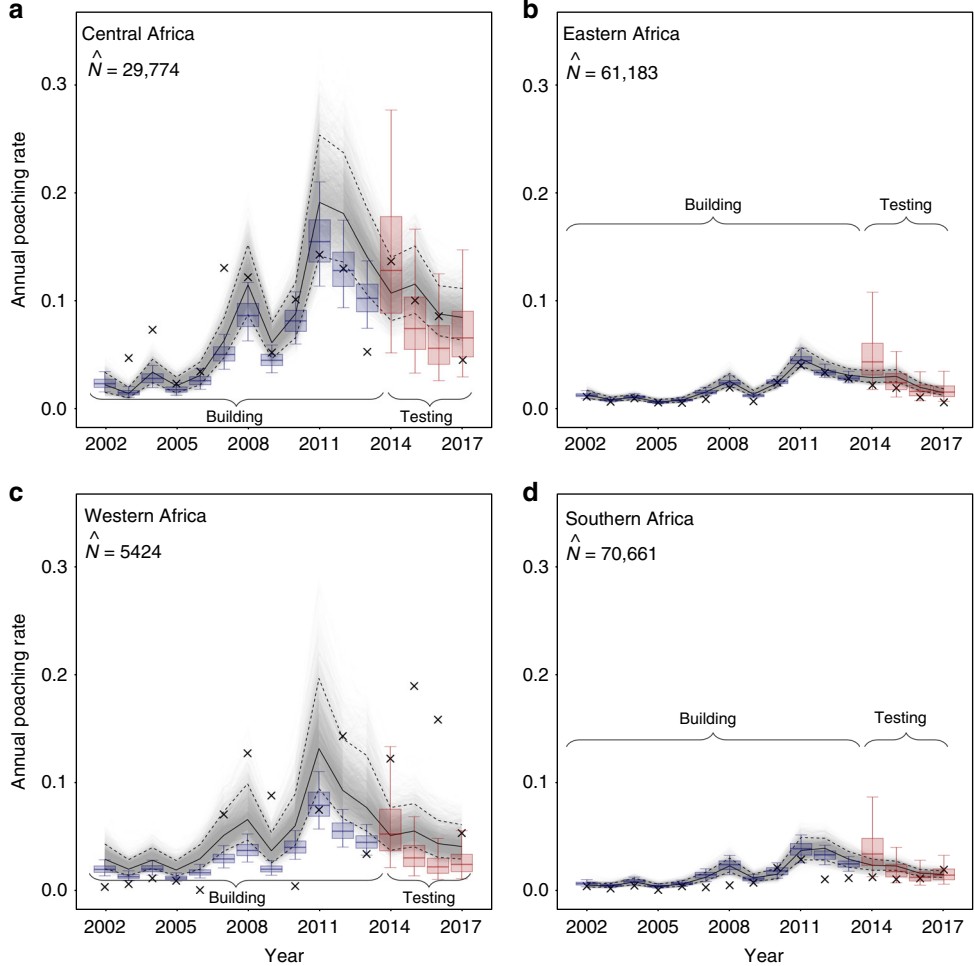

**Fig. 4** Region-specific annual estimates of per capita poaching rates. Displayed estimates are annual median poaching rates across the sites and years within **a** Central Africa, **b** Eastern Africa, **c** Western Africa and **d** Southern Africa, each derived from 3000 MCMC samples. Grey lines display estimates from the model fitted to all data, with solid black line showing median estimates and dashed lines outlining 90% CIs. Blue boxes represent estimates from the model (same structure) fitted to training data (2002–13) only, red colour highlights estimates for test data (2014–17). Boxplot centre lines represent median estimates, box bounds first and third quartiles, and whiskers 90% CIs. Crosses represent overall observed poaching rate across the sites within the respective region and will be biased towards sites where more carcasses are found. $\hat{N}$ represents the sum of the latest population size estimates over all sites within the respective region[15]

compared to other regions (cf. Supplementary Results, pages 5, 20, 31 and 47).

It is striking that poverty and corruption-related covariates correlate with local and country-wide levels of poaching more strongly than estimates of law enforcement adequacy (cf. Fig. 3a–f). This is possibly due to difficulty in estimating law enforcement adequacy; however, all assessors were trained and we consider it likely that targeting poverty and corruption really are more effective options. Tackling corruption is difficult, but conservation has started working in this area[40]. Tackling poverty is also notoriously difficult, but there has been a recent increase in attempts to link conservation to poverty alleviation[41]. It is important to note that the association we find with poverty is not specific to community-based natural resource management programmes: we find a generally lower poaching rate in areas where multiple indicators of poverty are lower and we have not looked specifically at whether conservation at a site has significant poverty alleviation goals. If well implemented, community-based conservation efforts that combine wildlife conservation with material benefits to community members may improve both conservation status and reduce poverty[42,43]. Notable successes for elephant conservation include the example of Namibian

conservancies where elephant poaching has been substantially reduced[44], though impacts on poverty are mixed[41,45], but these good examples should be read within a context of highly variable impacts on poaching in community-based natural resource management systems[46]. Our results support both community-based conservation projects where they provide genuine poverty-alleviation and increased involvement by conservation in more general poverty reduction schemes. However, as both corruption and poverty are recalcitrant problems, focusing conservation activity purely on reducing supply is unwise.

In conclusion, we confirm that elephant poaching rates in Africa have started to decline after the peak in 2011, reaching plausible natural birth rates again in 2016/2017. As natural mortality is likely not entirely compensatory to poaching deaths, Africa's elephant populations remain threatened without continued reductions in poaching. Interventions need to focus both on reducing demand and supply of ivory: as ivory prices rise, demand seems to change relatively little[47], but our results suggest supply changes strongly. Any reduction in demand may therefore markedly reduce elephant poaching rates. While the observed changes in demand are consistent with the signals sent by Chinas elephant ivory trade ban and demand reduction campaigns by

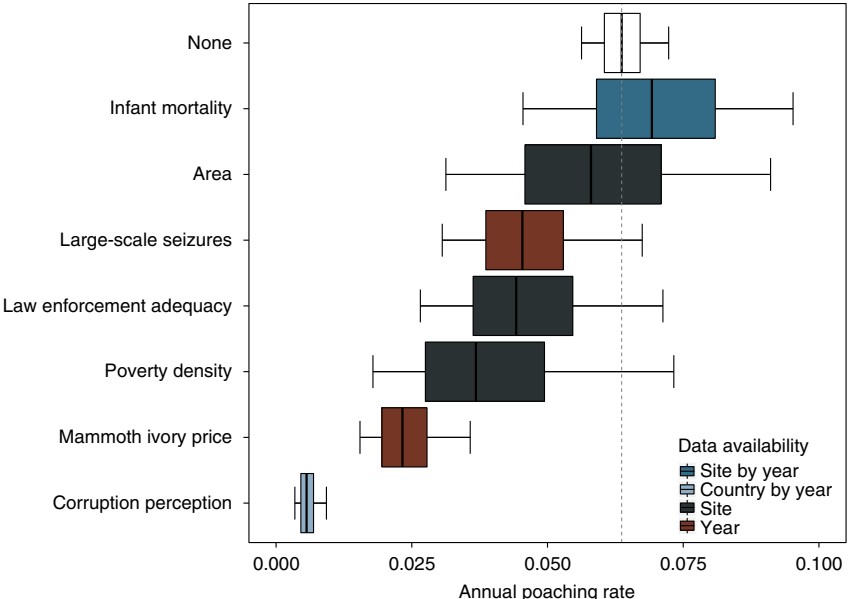

**Fig. 5** Sensitivity of estimated poaching rates to different simulated conservation targets. Each box represents 3000 median MCMC samples from model predictions between 2006 and 2017, where each covariate in turn is set to the best value at any site, country or year, while the other covariates were set to their observed values. The white box displays poaching rate median estimates without simulated intervention. Boxplot centre lines represent median estimates, box bounds first and third quartiles, and whiskers 90% CIs. For predictions, precipitation was set to the average value across sites and years to reduce variation in natural mortality rates across sites and years. The colour scheme is the same as in Fig. 3

NGOs[9,17], falling demand might also be a result of a Chinese economic slowdown[48]. Finally, we suggest that improving law enforcement using conventional methods in many areas might reduce elephant poaching, but reductions in poverty and corruption in communities neighbouring protected areas may have a greater effect and obvious additional benefits.

## Methods

**Elephant carcass encounter data**. African elephant carcass data were collected as part of the Monitoring the Illegal Killing of Elephants (MIKE) programme, which was instituted by the Convention on International Trade in Endangered Species (CITES) in 2002 and since then has worked with wildlife authorities across Africa to implement a ranger-based monitoring programme. The programme collates annual carcass counts from 53 sites (mostly protected areas, but often extending into neighbouring unprotected zones) in 29 countries across sub-saharan Africa. Full details of the monitoring methods are described elsewhere[12], but, in essence, rangers on regular patrols record the location of any elephant carcass encountered and identify whether death was the result of natural mortality, management or illegal killing (almost always poaching for ivory, but very occasionally the result of retaliation in human-elephant conflict). Between 2002 and 2017, the programme has recorded 18,007 carcasses in Africa, of which 8860 were identified as illegal killings, providing 607 observations from 53 sites in 16 years (includes all records received by February 2018). Several sites did not report carcasses every year, or joined the programme later than 2002.

It should be noted that these carcass encounter data, collated by the MIKE programme, show a few potential limitations[12]: (a) variation in background mortality (i.e. carcasses resulting from natural mortality or management) is unknown, but influences PIKE as it is assumed to be constant across years and sites. Background mortality (here natural mortality) is increased during droughts and periods of low rainfall[49,50], so we aimed to account for variable natural mortality by estimating the effect of site-specific annual precipitation on PIKE and setting this effect to zero for the predictions of the model. (b) Calculating PIKE across sites and years is based on the assumption that detection probabilities are the same for all carcasses, resulting from illegal activities, management or natural reasons. This might be an unlikely assumption, as the data are collected by anti-poaching patrols with the objective to deter illegal activities. It seems plausible to assume, however, that this bias is rather constant across space and time leading to an accurate estimation of trends and association with covariates. (c) Based on data from 53 sites across Africa, the prediction of poaching rates might not cover the full uncertainty of continental estimates, yet the surveyed area covers 25% of the area where African savannah elephants are extant residents[51] and about 50% of Africa's savannah elephant population[6,15].

**Covariates**. The choice of covariates (Supplementary Table 1), considered as potential drivers of poaching intensity, was guided by previous studies[7,12] and expert knowledge[52]. We included covariates that we considered to relate to demand or supply of elephant ivory, including factors that vary at temporal and spatial levels and two separate indicators of poverty: infant mortality rate and poverty density. Poverty is a complex, multidimensional problem that cannot easily be measured in a single variable[53], but the negative impact of poverty on illegal wildlife activities has been highlighted before[18] so it is important to consider multiple aspects of poverty. Not all covariates were available at the highest site-by-year resolution. Below, we present them in the following order: site-by-year, country-by-year, site-level, annual. Before the analysis, all covariates were centred and standardised to have a mean of 0 and a standard deviation of 1. We also tested for collinearity among predictors. All combinations showed Spearman's $\rho^2$ estimates <0.5, which we considered a non-problematic correlation (see Supplementary Fig. 1).

Infant mortality rate: The infant mortality rate (IMR) measures the number of deaths of children under one year of age per 1000 live births and is a crude indicator of development and socio-economic status levels in a community[54]. Note that IMR is included entirely as a proxy for one axis of poverty[55]: if IMR is strongly predictive of elephant poaching rates we would not interpret this as suggesting that healthcare interventions alone would be expected to impact poaching rates.

IMR estimates were available at site level for the year 2000, produced by the Centre for International Earth Science Information Network (CIESIN)[56]. Annual IMR estimates by country were made available by the United Nations (UN) inter-agency group for child mortality estimation[57]. As both spatial and temporal variability are high, we combined the two datasets to obtain IMR estimates for every site in every year. In reality, improvement rates in rural and urban areas may differ, but national changes likely reflect greater improvements in the rural areas where elephant populations are most common and IMR is higher, rather than smaller changes from urbanisation[58].

It is important to note that spatial differences in average IMR might represent differences between sites in poverty better than annual IMR measures. Annual IMR declines over time as successful medical and public health measures have improved healthcare much faster than other factors associated with poverty have improved, potentially weakening the value of annual IMR as a proxy for overall poverty. We therefore tested the correlation of site-level IMR[56] with PIKE in a separate model, in which we neglected the temporal changes in IMR entirely. The results of this supplementary analysis supports the assumption that spatial variation in IMR is a better indicator of poverty than temporal variation (see Supplementary Table 5).

Precipitation: Annual precipitation per site was derived from the Climate Hazards Group InfraRed Precipitation with Station data (CHIRPS)[59]. In the analysis, we took the natural logarithm of this variable because of its left skewed distribution. This climate variable was included to allow for changes in natural elephant mortality. Variation might arise from two processes. Sites with higher precipitation may identify denser habitats, where finding carcasses due to natural

mortality is more difficult, and hence PIKE may be higher due to underestimated natural mortality. Secondly, lower precipitation (within or among sites) may increase natural mortality[49,50] and thus lead to underestimated poaching rates because of lower PIKE values.

Corruption perceptions index: Corruption perceptions index (CPI) was derived by Transparency International[60] for every country in every year. It represents the perceived level of public sector corruption of a country according to experts and businesspeople. The index uses a scale of 0 to 100, where 0 is 'highly corrupt' and 100 is 'very clean'. We included CPI as a proxy for public sector and political corruption, which has been shown to affect the presence of illegal wildlife activities[27].

Poverty density: Poverty density defines the number of people per km² earning less than US$ 1.25 per day. It represents a measure of relative poverty and thus another proxy of the multidimensional poverty problem. These site level data were provided for the year 2005 by HarvestChoice[61].

Site area: Surface area of MIKE sites[62] in km². In the analysis, we took the natural logarithm of this variable because of its left skewed distribution. The expected effect of the site area on poaching intensity is somewhat ambivalent. On the one hand, larger protected areas might exhibit less of the negative edge effect, on the other hand, smaller ecosystems might be easier to patrol.

Law enforcement adequacy: Estimates of the adequacy of law enforcement provision. For each site, MIKE specialists return a form after receiving training from (https://cites.org/eng/prog/mike/tools_training_materials/leca) the MIKE programme team, estimating the adequacy of law enforcement provision. We expected ecosystems with higher law enforcement adequacy to show lower PIKE values.

Large-scale ivory seizures: Annual weight of large scale ivory seizures (≥500 kg)[63,64]. In cases, where worked ivory was part of the consignment, the values were converted to raw ivory equivalent, factoring in a 30% loss during processing.

Ivory price: Annual mammoth ivory prices in the main Chinese markets (China, Hong Kong and Macao) were derived from the UN Comtrade database[65]. This covariate was included as a proxy for demand for elephant ivory, as we assume that mammoth ivory prices are correlated with black market elephant ivory prices (for which data do not exist). Yet, it is worth noting that price for ivory is likely not only affected by market demand, but also more general conditions of the economy. To correct the obtained trade values for varying inflation rates, we used World Bank consumer price indices[66]. The corrected trade values were averaged by taking the market specific net weight into account. Note that Macao only reported mammoth ivory prices for the years 2006–09 and 2014.

Ivory price and impacts of seizures on supply and demand may influence poaching rates over a variety of time-scales. While poachers in Africa may be aware of international trends, it is possible information about markets flows slowly to the field. Consequently, we repeated all our analyses with lags of up to two years in these two variables, as is common within econometric analyses[67]. In the main results we present the zero-lag model.

**Statistical analysis**. Inferring elephant poaching intensity from carcass encounter data is difficult when, as here, sampling effort is unknown. Estimating the proportion of illegally killed elephants (PIKE), a relative measure, somewhat reduces this issue, assuming sampling effort is invariant for carcasses of natural and illegal causes in a particular year and site.

To estimate PIKE for each observation $i$, we assumed the number of carcasses identified as illegal killings ($n_{illegal}$) to be a binomial random variable given the total number of elephant carcasses ($n_{total}$) and probability $p$, such that

$$n_{illegal,i} \sim \text{Binomial}(p_i, n_{total,i}), \tag{1}$$

where probability $p_i$ (=estimated PIKE) is a function of a set of a priori chosen environmental and socio-economic covariates and year-, country- and site-level normally distributed ($\mathcal{N}$) random intercepts with level-specific means ($\mu$) and standard deviations ($\sigma$), transformed using the canonical logit link:

$$\text{logit}(p_i) = \beta_0 + \beta_1 \ln(\text{Precip}_i) + \beta_2 \text{IMR}_i + \beta_3 \text{CPI}_{\text{country}[i]} \\ + \mathcal{N}(\mu_{\text{site}[i]}, \sigma_{\text{site}[i]}) + \mathcal{N}(\mu_{\text{year}[i]}, \sigma_{\text{year}[i]}) + \mathcal{N}(0, \sigma_{\text{country}[i]}). \tag{2}$$

To account for the spatial and temporal structure of the data, the hierarchical level means for site ($\mu_{\text{site}}$) and year ($\mu_{\text{year}}$) were modelled in detail such that

$$\mu_{\text{site},s} = \beta_4 \text{PovDens}_s + \beta_5 \text{LawEnf}_s + \beta_6 \ln(\text{Area})_s, \tag{3}$$

$$\mu_{\text{year},y} = \beta_7 \text{Seizures}_y + \beta_8 \text{IvoryPrice}_y. \tag{4}$$

$\beta_n$ represent the regression coefficients, CPI is the annual country-level corruption perceptions index, PovDens (poverty density), Area (site area) and LawEnf (law enforcement adequacy) are site-level covariates, Seizures (large scale ivory seizures) and IvoryPrice (ivory price) are annual-level covariates, and Precip (precipitation) and IMR (infant mortality rate) are annual site-level covariates.

The model was fitted via Markov Chain Monte Carlo (MCMC) sampling using the software JAGS[68], accessed through the R[69] package R2jags[70]. The parameter posterior estimates were derived in three independent chains, each of 100,000 iterations, a burn-in phase of 50,000 iterations and thinned to every 50th sample.

Based on estimated $\hat{R}$[71] and effective sample sizes the applied MCMC algorithm fully converged (see Supplementary Table 2).

With the objective to build an interpretable model with high predictive capacity of PIKE, we regularised the model using the Bayesian lasso[72] instead of applying subset selection. By imposing a penalty proportional to the absolute values of the regression coefficients (L₁-norm penalty), the lasso[73] automates variable selection using continuous shrinkage and leads to a sparse model representation. In Bayesian inference, we achieve this using Laplace priors for the regression parameters $\beta_n$, such that

$$\beta_n \sim \text{Laplace}(\mu = 0, b = \lambda^{-1}), \tag{5}$$

where the regularisation parameter, $\lambda$, represents the inverse of the scale parameter in the Laplace distribution (or the rate in an exponential distribution), resulting in stronger shrinkage with increasing $\lambda$. We allowed the model to estimate $\lambda$ from the data by setting it as a hyperparameter. For its implicit estimation, we imposed a diffuse gamma hyperprior on $\lambda^2$ to maintain conjugacy:

$$\lambda^2 \sim \frac{\delta^r}{\Gamma(r)}(\lambda^2)^{r-1} e^{\delta\lambda^2}, \tag{6}$$

with shape $r = 1$ and rate $\delta = 1$, which resulted in a median posterior estimate of $\lambda = 1.64$ (90% CI: 1.00–2.42). We also used gamma priors with $r = 1$ and $\delta = 1$ on the standard deviations of the year-, country- and site-level random effects. We tested the sensitivity of the choice of prior distributions on $\lambda$, $\sigma_{\text{site}}$, $\sigma_{\text{year}}$ and $\sigma_{\text{country}}$. The regression coefficients showed little difference when imposing uniform instead of gamma prior distributions (compare Supplementary Tables 2 and 3).

For an assessment of the models predictive capacity accounting for potential temporal dependencies[31], we first sliced the data into temporal blocks of training and test sets. Training data comprises all records in the period 2002 to 2013 ($n_{\text{training}} = 447$, i.e. ~75%). Test data are all observations between 2014 and 2017 ($n_{\text{test}} = 160$, i.e. ~25%). To validate the model, we estimated PIKE for the period 2014–17 from 3000 MCMC draws of the model fitted to the training data. These estimates were compared against the respective PIKE observations in the test set (Fig. 2b). As a measure of predictive power we calculated $R^2$ weighted by $n_{\text{total}}$

**Estimating annual poaching rates**. While the proportion of illegally killed elephants (PIKE) overcomes the problem of unknown sampling effort, the rate of illegal killing (i.e. the proportion of illegally killed elephants of total population) is more intuitive. Burnham[52] proposed a simple conversion from PIKE to poaching rate ($m_p$), given an pre-defined natural mortality rate ($m_n$):

$$m_p = \frac{\text{PIKE}\, m_n}{1 - \text{PIKE}} \tag{7}$$

As such, the derived poaching rate retains a perfect 1:1 relationship with PIKE. Based on estimates collated by Wittemyer et al.[7], we assumed a constant natural mortality of 3% ($m_n = 0.03$), but compared the results to estimates with 2% ($m_n = 0.02$) and 4% ($m_n = 0.04$) natural mortality (see Supplementary Fig. 3). It is worth noting that as PIKE tends to 1, the estimated poaching rate increases exponentially, which might lead to implausibly high poaching rates. Therefore, when estimating continental annual poaching rates, we depicted the median across site-specific annual poaching rates. Site-specific assessment (see Supplementary Results) was based on the estimated PIKE, because in some sites PIKE values were estimated close to 1. Note we did not impose a cap on estimated PIKE, as poaching rates even in large elephant populations can be extremely high[74].

To predict the annual continental poaching rate (grey lines in Fig. 2a), we drew 3000 samples from the posterior distribution to estimate PIKE for all surveyed sites and years, translated these into site-by-year poaching rates and took the annual median value. For the observed annual continental poaching rate (black crosses in Fig. 2a), we first summed up all observed carcasses across sites, derived annual continental PIKE values and turned them into annual continental poaching rates. Note that the latter might be biased downwards, because sites that report more carcasses (e.g. due to better resourced ranger patrols), and thus dominate the continentally aggregated PIKE observations, tend to have lower poaching rates than sites with fewer observations.

**Identifying conservation targets**. To identify potential conservation targets, we estimated the sensitivity of the estimated poaching rate to improvements in the socio-economic factors considered. We used 3000 MCMC samples from the fitted model to predict continental annual poaching rates (or region- and site-specific PIKE; see Supplementary Results) with the predictor values consecutively set to the best (i.e. most elephant friendly) observed value within all 53 sites and 15 years. These were: IMR = 17.73 deaths/1000 infants (Tarangire and Manyara National Parks, Tanzania 2016); CPI = 65 (Botswana 2012); poverty density = 4.85 people km⁻² at < US$ 1.25 per day (Lopé National Park, Gabon); site area = 81,046 km² (Selous and Mikumi National Parks, Tanzania); law enforcement adequacy = 0.83 (Etosha National Park, Namibia); large-scale ivory seizures = 790 kg (2002); mammoth ivory price = US$ 23.72 kg⁻¹ (2002). Thus, the differences among sites and countries (see Supplementary Results) are simply a consequence of the current situation in a site or country relative to the best situation in any site or country between 2002 and 2017, and do not represent different effect sizes among sites and countries.

**Spatial and temporal residual autocorrelation**. We checked for spatial and temporal residual autocorrelation using the Sncf function in the ncf R package[75], which allows for a spatio-temporally structured model to be analysed. Residuals were calculated as the difference of estimated and observed PIKE. We did not consider any of this further as the residuals showed neither consistent spatial nor temporal autocorrelation (Supplementary Fig. 2).

**Reporting summary**. Further information on research design is available in the Nature Research Reporting Summary linked to this article.

## Data availability

Data and Supplementary Results are available in a figshare data repository at https://doi.org/10.6084/m9.figshare.7713245.

## Code availability

The analysis of the model and all data manipulations were built in R v. 3.3[69]. The model was fitted in JAGS[68], accessed through the R package R2jags[70]. R code to reproduce the analysis is available in a figshare data repository at https://doi.org/10.6084/m9.figshare.7713245.

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

## Acknowledgements

We thank the wildlife authorities and NGOs who return MIKE data. We thank the MIKE and ETIS technical advisory group for their review of this work, and the comments from F. Underwood and C. Thouless. We thank J. Jehle for assisting with the data preparation. The research, under CITES/MIKE, is funded by the European Union and this publication was produced with the financial support of the European Union. Its contents are the sole responsibility of the authors and do not necessarily reflect the views of the European Union. C.M.B acknowledges support from NERC grant NE/N001370/1. The views expressed herein are those of the author(s) and do not necessarily reflect the views of the United Nations.

## Author contributions

S.H., J.B. and C.M.B. conceived the project. S.H. and C.M.B. undertook the analysis and wrote the manuscript. S.H. produced the figures. M.K., J.B. and C.F.D. contributed critically to the drafting and revision of this manuscript.

## Additional information

**Competing interests:** While conducting this research, M.K. and J.B. were employed by the MIKE programme. This does not alter our adherence to *Nature Communications'* policies on sharing data and materials. The authors declare no competing interests.

