## [Peer Review File · Nature Communications]

Reviewers' comments:

Reviewer #1 (Remarks to the Author):

This is an interesting and thought-provoking analysis which explores associations between site-level proportion of elephants which are illegally killed and a range of global, national and site-level covariates. It concludes that reducing corruption and tackling demand would be the most effective ways of reducing poaching rates.

The paper is mostly quite careful in its interpretations, and the underlying statistical methods are sound. However, there are a number of important points where I think the interpretation is oversold and needs to be more realistically expressed. The variables tested are only ever going to be imperfect proxies for complex processes which vary in time and between sites, and the relationships found are only associations. This needs to be made very clear throughout.

Title: I recognise "alongside" as a word that is trying to get away from implications of causation but instead it means that the title is both not very clearly expressed and still implying causation. Also, and importantly, the corruption perception index used is national, not local, and mammoth ivory price is not necessarily a strong proxy for ivory demand. I think the title instead should be more descriptive of the content of the paper, e.g. "Associations between estimated African elephant poaching rates and site-level and international factors hypothesised to incentivise poaching"

A major issue in attributing causation is the likely presence of lags in the relationship between variables. In particular the mammoth ivory price is apparently used in the model without a lag, which means that it is highly unlikely that it (or the unobservable variable of black market ivory price which it's proxying for) is affecting site-level poaching directly. It would be better to include it in the model with a range of lags as would be done in an econometric analysis.

Lines 106-108, Fig 3a,b. I think you should tone down the claims that these data provide "excellent" validation for the model, given that the fits are not particularly good.

Line 111. You therefore can't say "strong evidence that ivory price drove annual variation in poaching rates". This is not possible given the model structure.

Generally speaking the paper needs a thorough check for all instances where causative statements like this are made, which are unsupported (and unsupportable) by this type of analysis.

Line 114: "the logit-scale variance in the estimated site-level mean of 0.25 (eq. 3) was similar to temporal variation, suggesting local action might be effective at halting poaching". Please explain what you mean by this? It appears that what you mean is that there is quite a lot of variance in the site-level mean, which you take to mean that there's room for improvement at the site level if the right levers are pulled. But it could just mean that there's heterogeneity between sites which may or may not be influencable. I don't think this sentence helps, and it could be deleted.

Line 122. Re compensatory nature of poaching - there's little evidence either way, and you don't give a reference, so I don't think you can assume anything (you're implying that poaching is not compensatory for natural mortality). Black rhino example - I'm not sure it's true that even the most protected populations are suffering a high (implied unsustainable) level of poaching (and the reference is 5 years old).

Line 138, line 156 (and title): I think you need to be really careful making site-level inferences about the effects of a country-level variable (CPI).

Line 141-142. You need to be careful when you say what covariate influences things more,

because you are setting each covariate to the best possible value found in the dataset. For some covariates the best value will be large relative to the SD of the distribution of values, for others it will be smaller. So you're not comparing like with like. It's OK if you're careful with your language, but here you are making unfair comparisons.

Figure 4: Can you clarify whether precipitation was allowed to vary between sites and year in this analysis, or if it was just set to the average value as the legend seems to imply? If the latter, then it is not actually controlling at all for natural mortality, because the whole point is that natural mortality varies by site and year.

Line 303. A constant natural mortality over site and year is not a good assumption because it's clearly not the case (particularly as we know that over the timeseries certain sites have experienced periods of intense drought). The poaching rates that you infer when using this equation are therefore not going to be particularly reliable. I think focussing more on PIKE is more defensible.

Reviewer #2 (Remarks to the Author):

This piece of work has the potential to be a very influential paper once published. As such, I have made a number of comments that I hope will improve the readability and accessibility of the material, not only to experts in the field, but to other interested readers as well. Generally, I find the data and analyses used to address the question of drivers of elephant poaching to be strong; my feedback is largely intended to improve the clarity of the manuscript and to perhaps better situate it among existing literature and concepts. I do also have a few questions and comments on the analyses themselves; see below.

The paper is extremely brief, almost terse, and I've suggested a number of places below where it would be useful to expand and elaborate on specific concepts or results, since there appears to be plenty of space within word counts to do so. In particular it's not clear why the authors do not delve more into their site and region-specific results? It seems like much more could be extracted from this part of their work, which would be of great interest to academics and practitioners working in particular geographies.

Specific comments:

21: replace "causing variation in" with "driving"

48-50: value of elephants for tourism could also be highlighted (Naidoo et al. Nat Comms 2017)

68: "...prices rises IN ELEPHANT IVORY".

73-74: "drugs to rhino horn" seems a strange example gradient of illegal markets. Maybe better to split into something like "both ecological (e.g. rhino horn) and other (e.g. narcotics, arms, etc) markets". Or something like that.

78-80: It's worth also mentioning poaching rates may be highest where conservation incentives are lowest. E.g. where HEC is high, or where local communities do not see any financial benefits from conservation. We can't focus only on law enforcement, community conservation incentives must also be brought into the equation. See Biggs et al. Cons Biol 2017. UPDATE: I see later that you make the same point. But better to be brought up here as well.

83-84: "intensity of elephant poaching is differentially sensitive to various aspect of supply and demand". Cumbersome and unclear wording. Rephrase.

104-107: why the temporal split of the dataset, rather than just holding 30-40% back across all sites and years and estimating predictive accuracy?

108-109: I understand space and format constraints, but need to at least briefly introduce the variables use you are using in your model before jumping straight to their predictive power

111: POSITIVELY correlated with poverty and NEGATIVELY correlated with law enforcement

114-115: Expand on this a bit; important point.

120-122: Elaborate on this. Important point on compensatory mortality which is raised but then glossed over.

137-139: There is a wealth of country and region-specific results contained in the Supporting Information yet this barely warrants a mention in the paper. Surely more discussion and presentation of these very interesting results is merited?

145-147: Reference (32) is well out of date; see various papers by Biggs, Naidoo, Roe, etc that would be a better citation here for the fact that effective community-based conservation can provide gains for both wildlife and people.

141-150: This section deserves more elaboration, as it's a key and perhaps underappreciated and surprising result (if not to those who work in this area). Interventions that focus on anti-poaching efforts tend to resonate more with policy makers and the general public, probably because they are more direct and "showy". But the longer, harder work of CBNRM may ultimately be a more effective strategy in reducing poaching rates of elephants – this should be emphasized more here, with better reference to the literature (refs mentioned above), and more discussion of relative investments and perceived impacts of CBNRM vs anti-poaching patrols. Militarization / surveillance of anti-poaching efforts in conservation, with potential implications on local communities and their attitudes towards wildlife conservation, also useful to discuss (see Biggs et al papers).

159-163: Last sentence of the paper is a strange way to conclude, as the paper doesn't address the reasons for falling demand in China. It would be better to end with conservation recommendations and implications thereof, rather than speculate on reasons for falling demand.

Fig. 1 – Make explicit reference to the pie pieces rather than just solid vs opaque...it took me a few mins to realize what you were trying to convey here. Also, worth emphasizing in the legend that green colour is "good" (i.e. declining PIKE) for elephants; also wasn't apparent at first glance.

Fig 2 – There is a lot of scatter / unexplained variation here. This does not come across from the text. I would emphasize somewhere that although you do see the trends you note, there is still a lot variation across sites that cannot be explained by these variables.

Fig 3a – Model seems to mostly overestimate PIKE values. Any ideas on why? What are the implications for the analysis?

Fig 4 – Legend is a bit confusing; first line needs to be clearer in terms of what the figure is actually showing. For covariates that are not set to global best values, are they held at mean values? Or drawn from distribution? Specify.

Table 2 – replace all beta coefficients with a shortened variable name, then provide full variable name in legend as you've already done. This will improve readability of table. E.g. B1 = "Prec"; B4="Poverty", etc. Are these covariates standardized? It appears so, and if that's the case, say in Table legend.

166-176: Although these are the best data out there, it's worth mentioning limitations of MIKE data here. Carcass detectability, variation across sites in level of reporting performance, relatively small # of 50+ sites to estimate continental parameters, etc.

181: Spell out IMR.

259-271: A number of variables are never defined (μ , big N, δ , etc).

273-276: What software program was used to run the MCMC sampling?

293-295: Provide justification here for why you do this split. It's not clear what the rationale is.

328: I understand what you're trying to do here, but does changing area make sense? Area of PAs is immutable (mostly), so if you're trying to simulate changes in conservation interventions across the set of MIKE sites then I don't think this makes sense. The others are all variables that in theory at least could be changed. Suggest removing area from the analyses (i.e. setting at it's mean value for the purpose of simulations).

Response to reviewer comments – Manuscript number
NCOMMS-18-36895-T:
African elephant poaching declines alongside falling ivory demand
and local corruption

February 14, 2019

Dear Reviewers,

Below, we respond in detail to your comments and suggestions. In the following list, your comments are italicized; our response is printed in normal font following each comment. Please note that line numbers and other references (figures, tables, sections) printed in bold refer to the originally submitted manuscript. In our responses, we refer to the line numbers in the revised manuscript.

Response to Reviewer 1:

This is an interesting and thought-provoking analysis which explores associations between site-level proportion of elephants which are illegally killed and a range of global, national and site-level covariates. It concludes that reducing corruption and tackling demand would be the most effective ways of reducing poaching rates. The paper is mostly quite careful in its interpretations, and the underlying statistical methods are sound. However, there are a number of important points where I think the interpretation is oversold and needs to be more realistically expressed. The variables tested are only ever going to be imperfect proxies for complex processes which vary in time and between sites, and the relationships found are only associations. This needs to be made very clear throughout.

We thank reviewer 1 for recognising the importance of our work, and identifying several points for improvement. In particular, we have gone through the manuscript (and title) to ensure we are appropriately circumspect about interpretations of our results. These and other changes are listed and detailed below.

1. **Title:** *I recognise “alongside” as a word that is trying to get away from implications of causation but instead it means that the title is both not very clearly expressed and still implying causation. Also, and importantly, the corruption perception index used is national, not local, and mammoth ivory price is not necessarily a strong proxy for ivory demand. I think the title instead should be more descriptive of the content of the paper, e.g. “Associations between estimated African elephant poaching rates and site-level and international factors hypothesised to incentivise poaching”*

While we agree that the title for this paper could be less implicative, we would like to highlight the main results clearly. Therefore we have changed reviewer 1’s suggestions to ‘African elephant poaching rates correlate with local poverty, national corruption and global ivory price’. As such, we avoid the implicative ‘alongside’ but state the main results in a clear and descriptive manner.

2. *A major issue in attributing causation is the likely presence of lags in the relationship between variables. In particular the mammoth ivory price is apparently used in the model without a lag, which means that it is highly unlikely that it (or the unobservable variable of black market ivory price which it’s proxying*

for) is affecting site-level poaching directly. It would be better to include it in the model with a range of lags as would be done in an econometric analysis.

We added an analysis of the effects of time-lagged ivory price and large-scale ivory seizures on PIKE. The results showed no evidence that alternative temporal lags in ivory price or seizures would change our other results (Tables 6-13). Furthermore, we found that ivory prices with one year lag were correlated with PIKE but with smaller effect size than the zero-lag models. Lags of two years removed the effect of ivory prices on PIKE almost entirely. A correlation of PIKE with seizures at different lags was never supported in our models (Tables 6-13). These results suggest that temporal lags between market prices and poaching rates are themselves short. We added summary tables for all additional models (Tables 6-13) and communicated the results in lines 142-148. We also added a paragraph describing the analysis in the methods section in lines 343-347.

3. **Lines 106-108, Fig 3a,b:** *I think you should tone down the claims that these data provide “excellent” validation for the model, given that the fits are not particularly good.*

We removed ‘excellent’ from this sentence (line 127), but we are still confident in the quality of our model. We believe an R^2 of 0.49 derived from the model validation on independent hold-out data (see Fig. 3b) is rather good considering the coarse spatial and temporal resolution of the data.

4. **Line 111:** *You therefore can’t say “strong evidence that ivory price drove annual variation in poaching rates”. This is not possible given the model structure.*

We toned this down to ‘we found a strong positive association between ivory price and annual variation in poaching rates’ in lines 136-137.

5. *Generally speaking the paper needs a thorough check for all instances where causative statements like this are made, which are unsupported (and unsupported) by this type of analysis.*

After a thorough check made the following changes:

- For the title, we changed ‘African elephant poaching declines alongside falling ivory demand and local corruption’ to ‘Local poverty, national corruption and global ivory price **correlate with** African elephant poaching’.
- In lines 136-137, we changed ‘we found strong evidence that ivory price drove annual variation in poaching rates’ to ‘we found a strong **positive association** between ivory price and annual variation in poaching rates’.
- In lines 189-191, we changed ‘Overall poaching levels were sensitive to changes in indices of poverty, but were more strongly linked to corruption and changes in ivory price, whilst seizures and land area had minimal impacts’ to ‘Overall poaching levels **seemed** sensitive to changes in indices of poverty, but were more strongly linked to corruption and changes in ivory price, whilst seizures and site area **showed** minimal impacts’.
- In lines 200-201, we changed ‘It is striking that poverty and corruption-related covariates influence local levels of poaching more strongly than estimates of law enforcement adequacy’ to ‘It is striking that poverty and corruption-related covariates **correlate with** local and country-wide levels of poaching more strongly than estimates of law enforcement adequacy’.
- In lines 231-232, we removed the aside ‘which would add to recent evidence that demand reduction can be successful’.

6. **Line 114:** *“the logit-scale variance in the estimated site-level mean of 0.25 (eq. 3) was similar to temporal variation, suggesting local action might be effective at halting poaching. Please explain what you mean by this? It appears that what you mean is that there is quite a lot of variance in the site-level mean, which you take to mean that there’s room for improvement at the site level if the right levers are pulled. But it could just mean that there’s heterogeneity between sites which may or may not be influencable. I don’t think this sentence helps, and it could be deleted.*

As suggested we have removed this sentence (lines 248-150).

7. **Line 122:** *Re compensatory nature of poaching - there's little evidence either way, and you don't give a reference, so I don't think you can assume anything (you're implying that poaching is not compensatory for natural mortality). Black rhino example - I'm not sure it's true that even the most protected populations are suffering a high (implied unsustainable) level of poaching (and the reference is 5 years old).*

It is correct that there is little evidence whether poaching compensates for natural mortality, which we hope to have clarified with additional text in lines 155-161. Yet, we believe it is worth mentioning that the estimated poaching rates for 2016/2017 can only be interpreted as sustainable if poaching compensates entirely for natural mortality. In the added text (lines 158-161), we briefly discuss in which ways poaching might influence population dynamics other than as compensated or additive mortality.

8. **Line 138, line 156 (and title):** *I think you need to be really careful making site-level inferences about the effects of a country-level variable (CPI).*

We clarified that CPI is a country-level variable in the **title**, in the abstract in line 27 and in lines 191-194 (previously **line 138**). We also added 'country-wide' to line 201. In line 234 (previously **line 156**), we make a general statement as part of the final conclusions of our paper, where we suggest that reductions in poverty and corruption might lead to reduced poaching. Having clarified the spatial resolution of CPI before, we think it is not necessary to do this again in these final remarks.

9. **Line 141-142:** *You need to be careful when you say what covariate influences things more, because you are setting each covariate to the best possible value found in the dataset. For some covariates the best value will be large relative to the SD of the distribution of values, for others it will be smaller. So you're not comparing like with like. It's OK if you're careful with your language, but here you are making unfair comparisons.*

With the analysis presented in Fig. 4 and lines 180-198, we are interested in exploring the estimated response in poaching rates to extreme but plausible conservation interventions. In Fig. 4 we do not want to show effect sizes (or for example the effect of reducing or increasing all factors by 10%). These results are shown in Table 2 and Fig. 2. Instead we explore best-case conservation scenarios: while the effect size of law enforcement adequacy is slightly larger than that of poverty density, Fig. 4 shows that reducing poverty might in reality be a more efficacious conservation target. This is because the majority of sites have relatively high law enforcement adequacy (close to the observed maximum of 0.83 at Namibia's Etosha NP), whereas only a few sites could improve substantially in this area. For poverty density the opposite is the case: few sites show low poverty density and the majority of sites would substantially reduce poverty if they achieved the 'best' observed value. We have clarified this in the added text in this paragraph (lines 180-198).

10. **Figure 4:** *Can you clarify whether precipitation was allowed to vary between sites and year in this analysis, or if it was just set to the average value as the legend seems to imply? If the latter, then it is not actually controlling at all for natural mortality, because the whole point is that natural mortality varies by site and year.*

For the fitting of the model we used precipitation data varying by site and year to account for site-by-year variation in natural mortality rates. For the predictions shown in Fig. 4, we used the observed data for all covariates except the respective conservation target and precipitation. The latter was set to the mean observed value (here zero, because all covariates were z-transformed before the analysis). By removing the precipitation effect (proxy for natural mortality), we reduce the variation in estimated PIKE caused by variation in natural mortality rates, which we are not interested in for this analysis. We have clarified this issue with the textual changes in the caption of Fig. 4, and a separate explanation in the added paragraph on MIKE data problems in lines 249-255.

11. **Line 303:** *A constant natural mortality over site and year is not a good assumption because it's clearly not the case (particularly as we know that over the timeseries certain sites have experienced periods of*

intense drought). The poaching rates that you infer when using this equation are therefore not going to be particularly reliable. I think focussing more on PIKE is more defensible.

It is certainly true that natural mortality rates vary between sites and years. By fitting the effect of precipitation on PIKE we assume to approximately capture the variation in natural mortality rates, which we then correct for in the estimated PIKE values by setting the precipitation effect to zero. We are aware that this correction might not fully cover the variance in natural mortality, but we believe it is a valid approach to substantially reduce it. We have added a detailed description of this issue and our approach in the newly added paragraph on MIKE data problems in lines 249-255. We also mention the two processes likely causing the effect of precipitation (or water availability) on variation in natural (or background) mortality in lines 300-305.

As natural mortality rates for African elephants are uncertain, we additionally show poaching rate predictions with assumed natural mortality rates of 2% and 4% (see Supplementary Fig. 3)

Reviewer 1 suggests focusing on PIKE instead of poaching rates is more reliable. In fact, the estimated poaching rate (see Eqn. 7) has a perfect 1:1 relationship with PIKE which we have now clarified in line 402. We present annual poaching rates, because we believe it is a more intuitive measure of poaching intensity than PIKE, but this does not make our paper more or less reliable.

Response to Reviewer 2:

This piece of work has the potential to be a very influential paper once published. As such, I have made a number of comments that I hope will improve the readability and accessibility of the material, not only to experts in the field, but to other interested readers as well. Generally, I find the data and analyses used to address the question of drivers of elephant poaching to be strong; my feedback is largely intended to improve the clarity of the manuscript and to perhaps better situate it among existing literature and concepts. I do also have a few questions and comments on the analyses themselves; see below.

We extend our gratitude to reviewer 2, particularly for excellent suggestions on where to expand and clarify our results and discussion section.

1. *The paper is extremely brief, almost terse, and I've suggested a number of places below where it would be useful to expand and elaborate on specific concepts or results, since there appears to be plenty of space within word counts to do so. In particular it's not clear why the authors do not delve more into their site and region-specific results? It seems like much more could be extracted from this part of their work, which would be of great interest to academics and practitioners working in particular geographies.*

Based on these suggestions we have expanded our manuscript (see below). In particular, we added a paragraph (see lines 168-178) to present region-specific trend analyses, which is supplemented by the new Fig. 5. Here we show the trend is very similar for all regions, but estimated poaching rates were substantially higher in central and western Africa. We also further discussed the ranked efficacy of conservation interventions on a regional level.

2. **Line 21:** *replace “causing variation in” with “driving”*
Done. See line 23.
3. **Lines 48-50:** *value of elephants for tourism could also be highlighted (Naidoo et al. Nat Comms 2017)*
Done. See line 54.
4. **Line 68:** *“prices rises IN ELEPHANT IVORY”.*
Done. See line 76.
5. **Lines 73-74:** *“drugs to rhino horn” seems a strange example gradient of illegal markets. Maybe better to split into something like “both ecological (e.g. rhino horn) and other (e.g. narcotics, arms, etc) markets”. Or something like that.*

Rephrased to ‘...thriving illegal markets persisting for natural (e.g. rhino horn) and other (e.g. narcotics, arms, etc.) products...’. See lines 81-82.

6. **Lines 78-80:** *It’s worth also mentioning poaching rates may be highest where conservation incentives are lowest. E.g. where HEC is high, or where local communities do not see any financial benefits from conservation. We can’t focus only on law enforcement, community conservation incentives must also be brought into the equation. See Biggs et al. Cons Biol 2017. UPDATE: I see later that you make the same point. But better to be brought up here as well.*

In lines 88-90, we have now briefly introduced the link between poverty reduction, community-based conservation and poaching intensity. Later on, in lines 203-221, we now elaborate further on the potential of CBNRM for local communities and elephant populations.

7. **Lines 83-84:** *“intensity of elephant poaching is differentially sensitive to various aspect of supply and demand”. Cumbersome and unclear wording. Rephrase.*

Rephrased to ‘Although it is plausible that elephant poaching cannot be halted without interventions aimed at multiple stages of the ivory trade, it is likely that elephant poaching responds more strongly to certain interventions than others.’. See lines 92-95.

8. **Lines 104-107:** *why the temporal split of the dataset, rather than just holding 30-40% back across all sites and years and estimating predictive accuracy?*

When validating a statistical model on data not used in the calibration of the model, the gold-standard is when hold-out data are structurally independent of the data used to fit the model. Roberts *et al.* (2017 *Ecography* 40:913-929) reviewed the concept of block cross-validation as an approach to independently validate such models. Here, we apply such block-validation technique to obtain more realistic validation statistics. We added an explanation why we validated our model on a temporal block of hold-out data and referred to Roberts *et al.* (2017 *Ecography* 40:913-929) in lines 390-391.

For the reviewers we briefly illustrate the importance of a validation on independent hold-out data: when randomly splitting our data into training and test set we obtain \widetilde{R}^2 of 0.54 (90% CI: 0.51-0.57). Compared to an \widetilde{R}^2 of 0.48 (90% CI: 0.39-0.54) resulting from a temporally blocked model validation, the random validation is too optimistic.

9. **Lines 108-109:** *I understand space and format constraints, but need to at least briefly introduce the variables use you are using in your model before jumping straight to their predictive power*

Done. See lines 114-120.

10. **Line 111:** *POSITIVELY correlated with poverty and NEGATIVELY correlated with law enforcement*

Done. See lines 137-139.

11. **Lines 114-115:** *Expand on this a bit; important point.*

We concur with reviewer 1 who suggested in her/his 6th comment to remove this sentence entirely. We intended to show that the amount of variance explained of site-level covariates is similar to the amount of variance explained by the global, annually varying variables to suggest that local action can be effective. We agree with reviewer 1 that comparing variances in site-level and year-level hierarchical means is not sufficient for such an interpretation and consequently removed this section.

12. **Lines 120-122:** *Elaborate on this. Important point on compensatory mortality which is raised but then glossed over.*

While there is little evidence about the compensatory nature of poaching mortality, we agree that it is worth mentioning that the estimated poaching rates for 2016/2017 can only be seen as sustainable if poaching compensates entirely for natural mortality. In the added text (lines 155-161), we also briefly discuss in which other ways poaching potentially influences the dynamics of African elephant populations. See also the response to reviewer 1’s 7th comment.

13. **Lines 137-139:** *There is a wealth of country and region-specific results contained in the Supporting Information yet this barely warrants a mention in the paper. Surely more discussion and presentation of these very interesting results is merited?*

While we agree with reviewer 2 that the supplementary information contains interesting site-, country- and region-specific results, we believe that the main focus should be on the continental analysis. Nevertheless, we added one paragraph (see lines 168-178) presenting poaching trends by region and discussing potential implications for conservation prioritization. We also made a clearer reference to the site-, country- and region-specific results regarding the efficacy of feasible conservation targets (see lines 191-198)

14. **Lines 145-147:** *Reference (32) is well out of date; see various papers by Biggs, Naidoo, Roe, etc that would be a better citation here for the fact that effective community-based conservation can provide gains for both wildlife and people.*

We replaced reference 32 with Naidoo *et al.* (2011 *Env. Conservation* 38:445-453) and Cooney *et al.* (2017 *Cons. Letters* 10:367-374). See line 212.

15. **Lines 141-150:** *This section deserves more elaboration, as it's a key and perhaps underappreciated and surprising result (if not to those who work in this area). Interventions that focus on anti-poaching efforts tend to resonate more with policy makers and the general public, probably because they are more direct and "showy". But the longer, harder work of CBNRM may ultimately be a more effective strategy in reducing poaching rates of elephants – this should be emphasized more here, with better reference to the literature (refs mentioned above), and more discussion of relative investments and perceived impacts of CBNRM vs anti-poaching patrols. Militarization / surveillance of anti-poaching efforts in conservation, with potential implications on local communities and their attitudes towards wildlife conservation, also useful to discuss (see Biggs et al papers).*

We thank reviewer 2 for this suggestion. We have expanded in this section (in lines 200-219) elaborating on the potential of alleviating poverty and corruption as poaching interventions. In this context we also emphasized how such conservation efforts can be integrated in CBNRM. But it is important to note that our paper does not explicitly assess the value of CBNRM (only the value of having fewer poor people near protected areas) so we don't want to over-interpret these results. We also added examples of successful CBNRM and updated the references to existing literature.

16. **Lines 159-163:** *Last sentence of the paper is a strange way to conclude, as the paper doesn't address the reasons for falling demand in China. It would be better to end with conservation recommendations and implications thereof, rather than speculate on reasons for falling demand.*

Done. See lines 233-235.

17. **Figure 1:** *Make explicit reference to the pie pieces rather than just solid vs opaque it took me a few mins to realize what you were trying to convey here. Also, worth emphasizing in the legend that green colour is "good" (i.e. declining PIKE) for elephants; also wasn't apparent at first glance.*

Done. See caption of Fig. 1.

18. **Figure 2:** *There is a lot of scatter / unexplained variation here. This does not come across from the text. I would emphasize somewhere that although you do see the trends you note, there is still a lot variation across sites that cannot be explained by these variables.*

We added a sentence in lines 140-142 to communicate that there remains variance not explained by our model.

19. **Figure 3a:** *Model seems to mostly overestimate PIKE values. Any ideas on why? What are the implications for the analysis?*

We believe that the continentally aggregated observed PIKE values may be biased low rather than that our model overestimates PIKE. We argue that sites with high numbers of reported carcasses

dominate the continental PIKE aggregations. These sites find more carcasses because they tend to be better resourced and thus have lower poaching rates than sites with fewer observations. This is only problematic for the aggregation of PIKE values over a large spatial extent, yet not for the analysis in general, assuming that sampling efforts are similar for natural deaths and carcasses resulting from illegal activity. We added this explanation in lines 127-132.

20. **Figure 4:** *Legend is a bit confusing; first line needs to be clearer in terms of what the figure is actually showing. For covariates that are not set to global best values, are they held at mean values? Or drawn from distribution? Specify.*

We changed the caption for Fig. 4 and added text to explain that we used the observed values for the other covariates. We hope these changes help to clarify what Fig. 4 is showing. Additional changes were made in lines 180-186.

21. **Table 2:** *replace all beta coefficients with a shortened variable name, then provide full variable name in legend as you've already done. This will improve readability of table. E.g. B1 = "Prec"; B4 = "Poverty", etc. Are these covariates standardized? It appears so, and if that's the case, say in Table legend.*

Done. See all table captions.

22. **Lines 166-176:** *Although these are the best data out there, it's worth mentioning limitations of MIKE data here. Carcass detectability, variation across sites in level of reporting performance, relatively small # of 50+ sites to estimate continental parameters, etc.*

We added a section on problems with MIKE data in lines 249-263 and referred to Burn *et al.* (2011 *PLOS ONE* 6:e24165) for additional information on these data.

23. **Line 181:** *Spell out IMR.*

Done. See line 268.

24. **Lines 259-271:** *A number of variables are never defined (μ , big N, delta, etc).*

Done. See lines 357, 358 and 285.

25. **Lines 273-276:** *What software program was used to run the MCMC sampling?*

We used JAGS for MCMC sampling. We added this information in lines 369-370.

26. **Lines 293-295:** *Provide justification here for why you do this split. It's not clear what the rationale is.*

We added an explanation why we validated our model on a temporal block of hold-out data and referred to Roberts *et al.* (2017 *Ecography* 40:913-929) in lines 390-391.

27. **Line 328:** *I understand what you're trying to do here, but does changing area make sense? Area of PAs is immutable (mostly), so if you're trying to simulate changes in conservation interventions across the set of MIKE sites then I don't think this makes sense. The others are all variables that in theory at least could be changed. Suggest removing area from the analyses (i.e. setting at it's mean value for the purpose of simulations).*

If there was a correlation of PIKE with site area, the interpretation could be for example that larger (or smaller) areas are potentially easier to protect. But this seems not to be the case, which we believe is a result worth reporting. While site area might not be a conservation target for existing MIKE sites, it could be a factor to consider for newly protected areas. Therefore, we have kept site area as part of this analysis.

Additional changes:

- In lines 16-39, we changed the abstract to meet the word limit.
- In line 44, we added information we had to remove from the abstract, namely that elephant populations have declined ‘both inside and outside protected areas’.
- In lines 53-55, we rephrased the importance of elephants for ecosystems and ecotourism.
- In lines 109-112, we added a brief description of our analysis at the end of the introduction. We removed this information from lines 98-100.
- In line 113, we added the heading ‘Results and Discussion’.
- In lines 120, 123, 126, 369, 394 and 442, we changed ‘fit’ to ‘fitted’ to ensure consistency throughout the manuscript.
- In line 153, we changed ‘In the second part’ to ‘In later parts’.
- In line 223, we added ‘Africa’s’ to clarify which elephant populations are meant.
- In lines 222-227, we changed wording in the conclusions.
- In lines 327-330, we removed discussion of the unknown nature of time-lags in large-scale ivory seizures, as we added an analysis of such effects.
- In lines 337-340, we removed the description of how we imputed the missing consumer price index for 2017, as these data are now available.
- We updated all references to the supplementary information.
- We split main manuscript and supplementary files.

REVIEWERS' COMMENTS:

Reviewer #1 (Remarks to the Author):

I am happy with the changes made in response to my review, and therefore happy to recommend publication.

Reviewer #2 (Remarks to the Author):

The authors have done a very good job responding to my concerns (and apparently that of the other reviewer as well). Consequently I have no further comments to make, other than the authors should replace or add to ref. 14 with the following, which is a very direct and specific demonstration of the value of elephants for tourism in Africa:

Naidoo, R., Fisher, B., Manica, A. & Balmford, A. Estimating the economic losses to tourism in Africa from the illegal killing of elephants. *Nat Commun* 13379 doi: 10.1038/ncomms13379 (2016).

Response to referees – Manuscript number NCOMMS-18-36895B:
African elephant poaching rates correlate with local poverty,
national corruption and global ivory price

March 21, 2019

Dear Reviewers,

Reviewer #2 suggested to *replace or add to ref. 14 with the following, which is a very direct and specific demonstration of the value of elephants for tourism in Africa: Naidoo, R., Fisher, B., Manica, A. & Balmford, A. Estimating the economic losses to tourism in Africa from the illegal killing of elephants. Nat Commun 13379 doi: 10.1038/ncomms13379 (2016).*

We concur with this suggestion and replaced ref. 14 with the above named reference.

Thank you again for your considered assessment of our manuscript.

Best wishes,
Severin Hauenstein (for all authors)